# Semantic Membership Inference Attack against Large Language Models

**Hamid Mozaffari**
Oracle Labs
hamid.mozaffari@oracle.com

**Virendra J. Marathe**
Oracle Labs
virendra.marathe@oracle.com

## Abstract

Membership Inference Attacks (MIAs) determine whether a specific data point was included in the training set of a target model. In this paper, we introduce the Semantic Membership Inference Attack (SMIA), a novel approach that enhances MIA performance by leveraging the semantic content of inputs and their perturbations. SMIA trains a neural network to analyze the target model's behavior on perturbed inputs, effectively capturing variations in output probability distributions between members and non-members. We conduct comprehensive evaluations on the Pythia and GPT-Neo model families using the Wikipedia and MIMIR datasets. Our results show that SMIA significantly outperforms existing MIAs; for instance, for Wikipedia, SMIA achieves an AUC-ROC of 67.39% on Pythia-12B, compared to 58.90% by the second-best attack.

## 1 Introduction

Large Language Models (LLMs) appear to be effective learners of natural language structure and patterns of its usage. However, a key contributing factor to their success is their ability to memorize their training data, often in a verbatim fashion. This memorized data can be reproduced verbatim at inference time, which effectively serves the purpose of information retrieval. However, this regurgitation of training data is also at the heart of privacy concerns in LLMs. Previous works have shown that LLMs leak some of their training data at inference time (Carlini et al., 2022b;a; Jagannatha et al., 2021; Lehman et al., 2021; Mattern et al., 2023; Mireshghallah et al., 2022; Nasr et al., 2023) Membership Inference Attacks (MIAs) (Shokri et al., 2017; Carlini et al., 2022b;a; Zhang et al., 2023; Ippolito et al., 2022) aim to determine whether a specific data sample (e.g. sentence, paragraph, document) was part of the training set of a target machine learning model. MIAs serve as efficient tools to measure memorization in LLMs.

Existing approaches to measure memorization in LLMs have predominantly focused on verbatim memorization, which involves identifying exact sequences reproduced from the training data. However, given the complexity and richness of natural language, we believe this method falls short. Natural language can represent the same ideas or sensitive data in numerous forms, through different levels of indirection and associations. This power of natural language makes verbatim memorization metrics inadequate to address the more nuanced problem of measuring semantic memorization, where LLMs internalize and reproduce the essence or meaning of training data sequences, not just their exact wording.

Previous MIAs against LLMs has predominantly focused on classifying members and non-members by analyzing the probabilities assigned to input texts or their perturbations (Carlini et al., 2021; Mattern et al., 2023; Shi et al., 2023; Zhang et al., 2024). In contrast, we introduce the Semantic Membership Inference Attack (SMIA), the first MIA to leverage the semantic content of input texts to enhance performance. SMIA involves training a neural network to understand the distinct behaviors exhibited by the target model when processing members versus non-members.

Our central hypothesis is that perturbing the input of a target model will result in differential changes in its output probability distribution for members and non-members, contingent on the extent of semantic change distance. Crucially, this behavior is presumed to be learnable. To implement this, we train the SMIA model to discern how the target model's behavior varies with different degrees of semantic changes for members and non-members. Post-training, the model can classify a given text

sequence as a member or non-member by evaluating the semantic distance and the corresponding changes in the target model's behavior for the original input and its perturbations.

Figure 1 illustrates the pipeline of our proposed SMIA inference. The SMIA inference pipeline for a given text $x$ and a target model $T(.)$ includes four key steps: **(1) Neighbor Generation:** The target sequence is perturbed $n$ times by randomly masking different positions and filling them using a masking model, such as T5 (Raffel et al., 2020), to generate a neighbour dataset $\tilde{x}$ (similar to Mattern et al. (2023); Mitchell et al. (2023)). **(2) Semantic Embedding Calculation:** The semantic embeddings of the input text and its neighbours are computed by using an embedding model, such as Cohere Embedding model (Cohere, 2024). **(3) Loss Calculation:** The loss values of the target model for the input text and its neighbours are calculated. **(4) Membership Probability Estimation:** The trained SMIA model is then used to estimate the membership probabilities. These scores are averaged and compared against a predefined threshold to classify the input as a member or non-member.

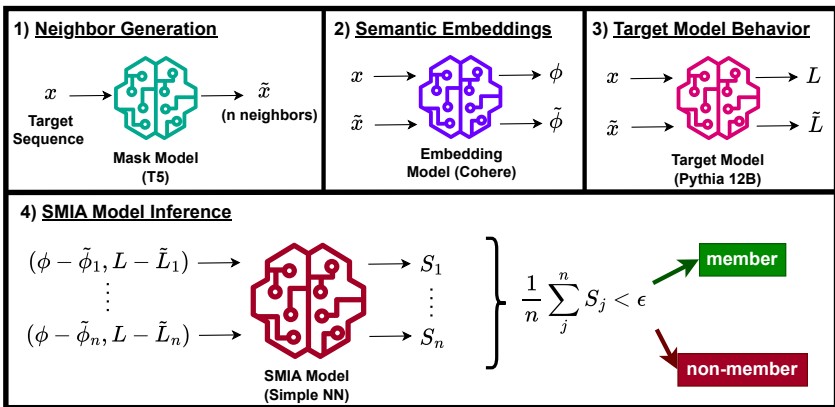

Figure 1: Our Semantic Membership Inference Attack (SMIA) inference pipeline.

**Empirical Results:** We evaluate the performance of our proposed SMIA across different model families, specifically Pythia and GPT-Neo, using the Wikipedia and MIMIR (Duan et al., 2024) datasets. To underscore the significance of the non-member dataset in evaluating MIAs, we include two distinct non-member datasets in our Wikipedia analysis: one derived from the exact distribution of the member dataset and another comprising Wikipedia pages published after a cutoff date, which exhibit lower n-gram similarity with the members. Additionally, we assess SMIA under two settings: (1) verbatim evaluation, where members exactly match the entries in the target training dataset, and (2) slightly modified members, where one word is either duplicated, added, or deleted from the original member data points.

Our results demonstrate that SMIA consistently outperforms all existing MIAs by a substantial margin. For instance, SMIA achieves an AUC-ROC of 67.39% for Pythia-12B on the Wikipedia dataset. In terms of True Positive Rate (TPR) at low False Positive Rate (FPR), SMIA achieves TPRs of 3.8% and 10.4% for 2% and 5% FPR, respectively, on the same model. In comparison, the second-best attack, the Reference attack, achieves an AUC-ROC of 58.90%, with TPRs of 1.1% and 6.7% for 2% and 5% FPR, respectively.

## 2 OUR PROPOSED SMIA

Membership inference attacks (MIAs) against LLMs aim to determine whether a given data point was part of the training dataset used to train the target model or not. Given a data point $x$ and a trained autoregressive model $T(.)$, which predicts $P(x_t|x_1, x_2, ..., x_{t-1})$ reflecting the likelihood of the sequence under the training data distribution, these attacks compute a membership score $A(x, T)$. By applying a threshold $\epsilon$ to this score, we can classify $x$ as a member (part of the training data) or a non-member. In Appendix B, we provide MIA use cases and details about how existing MIA work against LLMs.

MIAs seek to determine whether a specific data sample was part of the training set of a machine learning model, highlighting potential privacy risks associated with model training. Traditional MIAs typically verify if a text segment, ranging from a sentence to a full document, was used exactly as is in the training data. Such attacks tend to falter when minor modifications are made to the text, such as punctuation adjustments or word substitutions, while the overall meaning remains intact. We hypothesize that a LLM, having encountered specific content during training, will exhibit similar behaviors towards semantically similar text snippets during inference. Consequently, a LLM's response to semantically related inputs should display notable consistency.

In this paper, we introduce Semantic Membership Inference Attack (SMIA) against LLMs. This novel attack method enables an attacker to discern whether a *concept*, defined as a set of semantically akin token sequences, was part of the training data. Examples of such semantically linked concepts include "John Doe has leukemia" and "John Doe is undergoing chemotherapy." The proposed SMIA aims to capture a broader spectrum of data memorization incidents compared to traditional MIA, by determining whether the LLM was trained on any data encompassing the targeted concept.

Figure 2: Input features for our SMIA: semantic change and taregt model behaviour change for inputs and their neighbors.

## 2.1 SMIA Design

For the SMIA, we assume that the adversary has grey-box access to the target LLM, denoted as $T(.)$, which is trained on an unknown dataset $D_{\text{train}}$. The adversary can obtain loss values or log probabilities for any input text from this model, denoted as $\ell(., T)$, but lacks additional information such as model weights or gradients. The cornerstone of our SMIA is the distinguishable behavior modification exhibited by the target model when presented with semantic variants of member and non-member data points.

As illustrated in Figure 2, consider a 2-dimensional semantic space populated by data points. Members and non-members are represented by green circles and red circles, respectively. By generating semantic neighbors for both member and non-member data points (shown as green and red diamonds, respectively), we measure the semantic distance between targeted data points and their neighbors, denoted as $d_i^m$ and $d_i^n$. Subsequently, we observe the target model's response to these data points by assessing the differences in loss values (y-axis for log probability of that text under the taregt LLM data distribution), thereby training the SMIA to classify data points as members or non-members based on these observed patterns.

## 2.2 SMIA Pipeline

The SMIA consists of two primary components: initially, the adversary trains a neural network model $A(.)$ on a dataset gathered for this purpose, and subsequently uses this trained model for inference. The training and inference processes are detailed in Algorithms 1 and 2, respectively.

During the training phase, the adversary collects two distinct datasets: $D_{\text{tr-m}}$ (member dataset) and $D_{\text{tr-n}}$ (non-member dataset). $D_{\text{tr-m}}$ comprises texts known to be part of the training dataset of the target model $T(.)$, while $D_{\text{tr-n}}$ includes texts confirmed to be unseen by the target model during training. The adversary utilizes these datasets to develop a membership inference model capable of distinguishing between members ($\in D_{\text{tr-m}}$) and non-members ($\in D_{\text{tr-n}}$). For instance, Wikipedia articles or any publicly available data collected before a specified cutoff date are commonly part of many known datasets. Data collected after this cutoff date can be reliably assumed to be absent from the training datasets.

The SMIA training procedure, shown in Algorithm 1, involves four key stages:

---

**Algorithm 1** Our Proposed Semantic Membership Inference Attack: training

---

**Input**: dataset of members for training $D_{\text{tr-m}}$, dataset of non-members for training $D_{\text{tr-n}}$, masking model for neighbor generation $N(.)$, Embedding model $E(.)$, Target model $T(.)$, Number of neighbors $n$, Number of perturbations $k$, number of SMIA training epochs $R$, SMIA learning rate $r$, SMIA batch size $B$, loss function $\ell(.,.)$

**Output**: SMIA Model $A(.\,,.\,,D_{\text{tr-m}},D_{\text{tr-n}})$

1:  $D^m_{\text{masked}}, D^n_{\text{masked}} \leftarrow MASK(D_{\text{tr-m}}, n, k), MASK(D_{\text{tr-n}}, n, k)$      ▷ Masking
2:  $\tilde{D}^m, \tilde{D}^n \leftarrow N(D^m_{\text{masked}}), N(D^n_{\text{masked}})$      ▷ Neighbor generation
3:  $\Phi^m, \Phi^n, \tilde{\Phi}^m, \tilde{\Phi}^n \leftarrow E(D_{\text{tr-m}}), E(D_{\text{tr-n}}), E(\tilde{D}^m), E(\tilde{D}^n)$      ▷ Embedding
4:  $L^m, L^n, \tilde{L}^m, \tilde{L}^n \leftarrow \ell(D_{\text{tr-m}}, T), \ell(D_{\text{tr-n}}, T), \ell(\tilde{D}^m, T), \ell(\tilde{D}^n, T)$      ▷ Target model loss
5: Initialize $A(.)$
6: **for** e in $R$ **do**
7:      **for** $batch$ **do**
8:          **for** $i = 1$ **to** $B/2$ **do**
9:              $B^m \leftarrow (\Phi^m_{batch,i} - \tilde{\Phi}^m_{batch,i}, L^m_{batch,i} - \tilde{L}^m_{batch,i}, 1)$      ▷ Member half of the batch
10:          **end for**
11:          **for** $i = 1$ **to** $B/2$ **do**
12:              $B^n \leftarrow (\Phi^n_{batch,i} - \tilde{\Phi}^n_{batch,i}, L^n_{batch,i} - \tilde{L}^n_{batch,i}, 0)$ ▷ Non-Member half of the batch
13:          **end for**
14:          **update** $A(\{B^m, B^n\}, r)$      ▷ Update parameters of SMIA network
15:      **end for**
16: **end for**
17: **return** $A(.\,,.\,,D_{\text{tr-m}},D_{\text{tr-n}})$

---

**i) Neighbour generation (Algorithm 1 lines 1-2):**  The initial phase of SMIA involves generating a dataset of neighbours for both the member dataset ($D_{\text{tr-m}}$) and the non-member dataset ($D_{\text{tr-n}}$). The creation of a neighbour entails making changes to a data item that preserve most of its semantics and grammar, thereby ensuring that these neighbours are semantically closed to the original sample and should be assigned a highly similar likelihood under any textual probability distribution, as similar to Mattern et al. (2023); Mitchell et al. (2023). Specifically, Algorithm 1 line 1 describes the creation of masked versions of $D^m_{\text{masked}}$ and $D^n_{\text{masked}}$ by randomly replacing $k$ words within each text item $n$ times. Following this, in line 2, a Neighbour generator model $N(x, L, K)$—a masking model—is employed to refill these masked positions, generating datasets $\tilde{D}^m$ and $\tilde{D}^n$ for members and non-members, respectively. We utilize the T5 model (Raffel et al., 2020) in our experiments to perform these replacements, aiming to produce $n$ semantically close variants of each data point.

**ii) Calculate semantic embedding of the data points (Algorithm 1 line 3):**  The subsequent step involves computing semantic embeddings for both the original data points and their neighbours. As per Algorithm 1 line 3, we obtain the embedding vectors $\Phi^m \leftarrow E(D_{\text{tr-m}})$ and $\Phi^n \leftarrow E(D_{\text{tr-n}})$ for the member and non-member data points, respectively. Additionally, we calculate $\tilde{\Phi}^m \leftarrow E(\tilde{D}^m)$ and $\tilde{\Phi}^n \leftarrow E(\tilde{D}^n)$ for their respective neighbours. These vectors represent each data point's position in a semantic space encompassing all possible inputs. Our experiments leverage the Cohere Embedding V3 model (Cohere, 2024), which provides embeddings with 1024 dimensions, to capture these semantic features.

**iii) Monitor the behaviour of the target model for different inputs (Algorithm 1 line 4):**  The third stage entails monitoring the target model's response across data items in the four datasets. Here, we calculate the loss values: $L^m \leftarrow \ell(D_{\text{tr-m}}, T)$ for the member samples, $L^n \leftarrow \ell(D_{\text{tr-n}}, T)$ for the non-member samples, and similarly $\tilde{L}^m \leftarrow \ell(\tilde{D}^m, T)$ and $\tilde{L}^n \leftarrow \ell(\tilde{D}^n, T)$ for their respective neighbours. This step is crucial for understanding how the model's behavior varies between members and non-members under semantically equivalent perturbations.

**iv) Train an attack model (Algorithm 1 lines 5-16):**  The final phase of training involves developing a binary neural network capable of distinguishing between members and non-members by detecting patterns of semantic and behavioral changes induced by the perturbations. We initiate this by randomly initializing the attack model $A(.)$, then training it to discern differences between the semantic embeddings and loss values for each data point and its neighbours. The input features for

$A$ include the differences in semantic vectors $\Phi_i^m - \tilde{\Phi}_i^m$ and the changes in loss values $L_i^m - \tilde{L}_i^m$ for each sample $i$. Each sample is labeled '1' for members and '0' for non-members, with each training batch consisting of an equal mix of both, as suggested in prior research (Nasr et al., 2019). The model is trained over $R$ epochs using a learning rate $r$, culminating in a trained binary classifier that effectively distinguishes between members and non-members based on the observed data. We prvoide our SMIA training cost in Appendix C.

---

**Algorithm 2** Our Proposed Semantic Membership Inference Attack: inference

---

**Input**: Test input $x$, Trained SMIA Model $A(.\,,.\,,D_{\text{tr-m}}, D_{\text{tr-n}})$ on dataset of members for training $D_{\text{tr-m}}$ and dataset of non-members for training $D_{\text{tr-n}}$, masking model for neighbor generation $N(.)$, Embedding model $E(.)$, Target model $T(.)$, Number of neighbors in inference $n_{\text{inf}}$, Number of perturbations $k$, decision threshold $\epsilon$, loss function $\ell$

---

1: $x_{\text{masked}} \leftarrow MASK(x, n_{\text{inf}}, k)$          ▷ Masking
2: $\tilde{x} \leftarrow N(x_{\text{masked}})$          ▷ Neighbor generation
3: $\phi, \tilde{\phi} \leftarrow E(x), E(\tilde{x})$          ▷ Embedding
4: $L, \tilde{L} \leftarrow \ell(x, T), \ell(\tilde{x}, T)$          ▷ Target model loss
5: $\mu \leftarrow \frac{1}{n} \sum_{i \in [b]} A(\phi - \tilde{\phi}_i, L - \tilde{L}_i)$          ▷ Average of SMIA scores
6: **if** $\mu > \epsilon$ **then**
7:      **return** True          ▷ Member
8: **else**
9:      **return** False          ▷ Non-Member
10: **end if**

---

**SMIA Inference:** Upon completing the training of the model $A(.)$, it can be employed to assess whether a given input text $x$ was part of the target model $T(.)$'s training dataset. Algorithm 2 details the inference procedure, which mirrors the training process. Initially, $n_{\text{inf}}$ neighbours for $x$ are generated using the mask model (lines 1-2). Subsequently, we compute both the semantic embedding vectors and the loss values for $x$ and its neighbours $\tilde{x}$ (lines 3-4). These computed differences are then fed into the attack model $A(\phi - \tilde{\phi}_j, L - \tilde{L}_j)$, which evaluates each neighbour $j$. The final SMIA score for $x$ is determined by averaging the scores from all $n_{inf}$ neighbours (line 5), and this score is compared against a predefined threshold $\epsilon$ to ascertain membership or non-membership (line 6).

## 3 EXPERIMENT SETUP

In this section, we describe the models and datasets used in our experiments. Due to space constraints, we have organized additional information into appendices. We provide the details of the architecture for SMIA model in Appendix E.1, cost estimation of SMIA to Appendix C, privacy metrics used in our analysis in Appendix E.2, the hyperparameters for training the SMIA model in Appendix E.4, the baselines in Appendix B, and the computational resources utilized in Appendix E.5.

### 3.1 MODELS

**Target Models:** In our experiments, we evaluate our proposed SMIA across a diverse set of language models to assess its effectiveness and robustness. We utilize three categories of target models: (1) Pythia Model Suite: This category includes the largest models with 12B, 6.9B, and 2.7B parameters from the Pythia model suite (Biderman et al., 2023), trained on the Pile dataset (Gao et al., 2020). (2) Pythia-Deduped: It consists of models with the same parameterization (12B, 6.9B, and 2.7B) but trained on a deduplicated version of the Pile dataset. This variation allows us to analyze the impact of dataset deduplication on the effectiveness of MIAs. (3) GPT-Neo Family: To test the generality of our approach across different architectures, we include models from the GPT-NEO family (Black et al., 2021), specifically the 2.7B and 1.3B parameter models, also trained on the Pile dataset.

**Models Used in SMIA:** The SMIA framework incorporates three critical components: (1) Masking Model: We employ T5 with 3B parameters (Raffel et al., 2020) for generating perturbed versions of the texts, where random words are replaced to maintain semantic consistency. (2) Semantic Embedding Model: The Cohere Embedding V3 model (Cohere, 2024) is utilized to produce a 1024-dimensional semantic embedding vector for each text, enabling us to capture nuanced semantic variations. (3) Binary Neural Network Classifier: For the SMIA model, we utilize a relatively simple neural network (details shown in Table 8) with 1.2M parameters, which is trained to distinguish between member and non-member data points. In Appendix E.4, we discuss the hyperparameters that we use in our experiments for this model.

## 3.2 DATASETS

To evaluate the effectiveness of the SMIA, we need to collect three datasets: training dataset $D_{\text{tr}} = \{D_{\text{tr-m}}, D_{\text{tr-n}}\}$, validation dataset $D_{\text{val}} = \{D_{\text{val-m}}, D_{\text{val-n}}\}$, and test dataset $D_{\text{te}} = \{D_{\text{te-m}}, D_{\text{te-n}}\}$. Each dataset comprises a member and a non-member split. We employ the training dataset for model training, the validation dataset for tuning the hyperparameters, and the test dataset for evaluating the model performance based on various metrics.

### 3.2.1 WIKIPEDIA DATASET

**Wikipedia Training and Validation:** We selected a total of 14,000 samples from Wikipedia, verified as parts of the training or test split of the Pile dataset (Gao et al., 2020). This includes 7,000 member samples from the training split of the Wikipedia portion of Pile and 7,000 non-member samples from the test split. Samples were selected to have a minimum of 130 words and were truncated to a maximum of 150 words. Consistent with prior studies (Gao et al., 2020; Duan et al., 2024), we prepended article titles to the text of each article, separated by a "\n \n". The split for these samples assigns 6,000 from each category to the training dataset ($D_{\text{tr}}$) and 1,000 from each to the validation dataset ($D_{\text{val}}$). In Appendix C, we provide the cost estimation for preparing this dataset for our training. For example for Wikipedia training part, calculating the embedding vectors from Cohere model costs around \$32.

**Wikipedia Test:** For the test member dataset ($D_{\text{te-m}}$), we similarly sourced 1,000 samples from the training portion of Pile. Selecting an appropriate non-member dataset ($D_{\text{te-n}}$) for testing is crucial, as differences in data distribution between member and non-member samples can falsely influence the perceived success of membership inference. Prior research (Duan et al., 2024) indicates that non-member samples drawn from post-training publications or different sections of the Pile test dataset show varied overlap in linguistic features such as n-grams, which can affect inference results. To address this, we established two non-member test datasets: the first, referred to as Wikipedia Test ($WT = \{D_{\text{te-m}}, D_{\text{te-n}}^{\text{PileTest}}\}$), includes samples from Wikipedia pages before March 2020 that are part of the Pile test dataset. The second, called Wikipedia Cutoff ($WC = \{D_{\text{te-m}}, D_{\text{te-n}}^{\text{CutOff}}\}$), consists of 1,000 samples from Wikipedia pages published after August 2023, ensuring they were not part of the Pile training dataset.

### 3.2.2 MIMIR DATASET

The MIMIR dataset (Duan et al., 2024), a derivative of the Pile dataset (Gao et al., 2020), is designed to simulate real-world challenges in membership inference of LLMs. Members and non-members are drawn from the train and test splits of the Pile dataset respectively, with non-member samples designed to exhibit different n-gram overlaps. We specifically engaged with the most challenging MIMIR sub-split, where members and non-members share up to 80% overlap in 13-grams—a setting designed to rigorously test the discriminative power of our SMIA approach. We select Wikipedia_en, GitHub, PubMed Central, and ArXiv splits in our experiments. Samples were selected to have a minimum of 130 words. Each member and non-member dataset was then divided into 70% for training ($D_{\text{tr}}$), 10% for validation ($D_{\text{val}}$), and 20% for the test dataset ($D_{\text{te}}$). We benchmark the performance of SMIA against other baselines on the test datasets.

## 4 EXPERIMENTS

In this section, we present the experimental results of our SMIA and compare its performance to other MIAs in verbatim setting (for modified setting, see Appendix A). Due to space constraints, we

defer the TPR of attacks at low FPR to Appendix D.1, the effect of deduplication in the Pythia model family to Appendix D.2, analysis of SMIA's performance with varying numbers of neighbors during inference to Appendix D.3, the effect of training size on SMIA's performance to Appendix D.4, and, the histogram of similarities between generated neighbors and their original texts in both member and non-member training datasets to Appendix D.5.

## 4.1 Evaluation in Verbatim Setting

Our initial set of experiments aims to classify members and non-members without any modifications to the data, meaning that the members ($D_{\text{te-m}}$) in the test dataset are verbatim entries from the training dataset of the models. This evaluation setting is consistent with prior works (Yeom et al., 2018; Carlini et al., 2021; Shi et al., 2023; Mattern et al., 2023; Zhang et al., 2024). Table 1 and Table 2 present the AUC-ROC metric for various baseline methods and our proposed SMIA approach across different trained models for Wikipedia (with two distinct test datasets) and MIMIR dataset repectively (Refer to Appendix D.2 for evaluation results on deduplicated models). Additionally, Table 4 and Table 5 in Appendix D.1 provide the True Positive Rate (TPR) at low False Positive Rates (FPR) for these methods and datasets. For MIMIR experiments, the tables include the best AUC-ROC values for Min-K and Min-K++ across different values of $K$ and Pyhtia-1.4B as the reference model. The results demonstrate that SMIA significantly outperforms existing methods. For instance, on Pythia-12B and $WT = \{D_{\text{te-m}}, D_{\text{te-n}}^{\text{PileTest}}\}$ test dataset (i.e., when non-members are sampled from the same data distribution as members), SMIA achieves an AUC-ROC of 67.39% with TPRs of 3.8% and 10.4% at 2% and 5% FPR, respectively. In contrast, the LOSS method (Yeom et al., 2018) yields an AUC-ROC of 54.94% and TPRs of 2.1% and 5.8% at the same FPR thresholds. The Ref attack (Carlini et al., 2021), which utilizes Pythia 1.4B to determine the complexity of test data points on a reference model trained on the same data distribution (a challenging assumption in real-world scenarios), achieves an AUC-ROC of 58.90% with TPRs of 2% and 8.2% at 2% and 5% FPR. Furthermore, Min-K (Shi et al., 2023) and Min-K++ (Zhang et al., 2024) show better AUC-ROC compared to the LOSS attack, achieving 56.66% and 57.67% for $K = 20\%$.

On MIMIR dataset, SMIA demonstrates superior performance across multiple splits. For example, in the PubMed Central split, on Pythia-12B, it achieves an AUC-ROC of 68.39% with TPRs of 8.50%, 11.50%, and 30.50% at FPRs of 2%, 5% and 10%, respectively. The second-best attack, the Nei attack (Mattern et al., 2023; Mitchell et al., 2023), achieves a lower AUC-ROC of 57.77% with corresponding TPRs of 1.0%, 6.0%, and 12.50% at these FPR thresholds. Similar to previous work (Duan et al., 2024), we find Github split as a less challenging domain emphasizing that LLMs tend to memorize the code snippets with higher probability. These evaluations were conducted under the constraint of dataset sizes, with each split containing at most 1000 examples for members and 1000 examples for non-members (before splitting them into $\{D_{\text{tr}}, D_{\text{val}}, D_{\text{te}}\}$). It is important to note that these results are achieved with the constraint of a limited size for our training, validation, and test datasets. we postulate that with an expansion in the size of these datasets, SMIA would likely achieve even higher performance metrics.

**Why SMIA Outperforms Other MIAs:** SMIA delivers superior performance for two key reasons: Firstly, it incorporates the semantics of the input text into the analysis, unlike the baseline methods that solely rely on the target model's behavior (e.g., log probability) for their membership score calculations. Secondly, SMIA utilizes a neural network trained specifically to distinguish between members and non-members, offering a more dynamic and effective approach compared to the static statistical methods used by previous MIAs.

**Importance of non-member dataset:** In the other Wikipedia test dataset ($WC = \{D_{\text{te-m}}, D_{\text{te-n}}^{\text{CutOff}}\}$), where non-members are derived from Wikipedia pages published after August 2023, we observe a substantial improvement in SMIA performance, consistent with findings from other studies (Duan et al., 2024). For example, SMIA achieves an AUC-ROC of 67.39% and 93.35% for Pythia-12B on $WT$ and $WC$, respectively. In terms of TPR at low FPR for the same model, SMIA achieves 3.8% and 10.4% for 2% and 5% FPR with the $WT$ dataset, while achieving 46.2% and 66.0% for 2% and 5% FPR with the $WC$ dataset. This increase is also observed in other attack methods. For instance, Min-K++ (with $K = 10\%$) attains 54.77% AUC-ROC for the $WT$ dataset and 76.17% for the $WC$ dataset. The underlying reason for this is that the member dataset ($D_{\text{te-m}}$) has a higher n-gram overlap with the $WT$ non-member dataset compared to the $WC$ non-member dataset. A high n-gram overlap between members and non-members implies that substrings of non-

Table 1: AUC-ROC performance metrics for various MIAs, including our SMIA, evaluated on different trained models (Pythia and GPT-Neo) using the Wikipedia. The table compares results for verbatim member data $D_{\text{te-m}}$ entries against non-member datasets $D_{\text{te-n}}^{\text{PileTest}}$ and $D_{\text{te-n}}^{\text{CutOff}}$.

| Method | Pythia-12B | | Pythia-6.9B | | Pythia-2.7B | | GPT-Neo2.7B | | GPT-Neo1.3B | |
|---|---|---|---|---|---|---|---|---|---|---|
| | WT | WC | WT | WC | WT | WC | WT | WC | WT | WC |
| LOSS | 54.94 | 67.56 | 54.23 | 65.95 | 53.14 | 63.99 | 53.32 | 63.34 | 52.98 | 62.10 |
| Ref (Pythia 70M) | 52.73 | 58.29 | 51.71 | 56.42 | 49.92 | 53.74 | 50.07 | 53.86 | 49.70 | 52.91 |
| Ref (Pythia 1.4B) | 58.90 | 67.44 | 57.01 | 63.79 | 51.39 | 56.06 | 52.27 | 56.80 | 50.03 | 50.98 |
| Zlib | 54.33 | 66.56 | 53.61 | 64.98 | 52.54 | 63.01 | 52.70 | 62.59 | 52.42 | 61.38 |
| Nei | 55.83 | 72.06 | 55.17 | 70.78 | 53.87 | 69.13 | 53.51 | 68.34 | 53.08 | 67.36 |
| Min-K ($K = 10\%$) | 56.96 | 76.05 | 56.00 | 73.96 | 54.05 | 71.21 | 53.72 | 70.53 | 53.32 | 68.40 |
| Min-K ($K = 20\%$) | 56.66 | 73.90 | 55.65 | 71.95 | 53.86 | 69.26 | 53.66 | 68.68 | 53.36 | 66.82 |
| Min-K ($K = 30\%$) | 56.17 | 72.18 | 55.23 | 70.32 | 53.67 | 67.84 | 53.59 | 67.26 | 53.33 | 65.54 |
| Min-K++ ($K = 10\%$) | 56.83 | 78.47 | 54.77 | 76.17 | 52.37 | 72.38 | 51.73 | 72.93 | 51.57 | 69.87 |
| Min-K++ ($K = 20\%$) | 57.67 | 79.34 | 55.62 | 76.77 | 53.28 | 72.82 | 52.82 | 73.07 | 52.02 | 70.13 |
| Min-K++ ($K = 30\%$) | 57.76 | 78.96 | 55.81 | 76.21 | 53.62 | 72.27 | 53.21 | 72.46 | 52.41 | 69.52 |
| Our SMIA | **67.39** | **93.35** | **64.63** | **92.11** | **60.65** | **89.97** | **59.71** | **89.59** | **58.92** | **87.43** |

Table 2: AUC-ROC results for different MIAs on datasets in MIMIR dataset (Duan et al., 2024) where members and non-members share less than 80% overlap in 13-gram.

| Target Model | Dataset | Method | | | | | | |
|---|---|---|---|---|---|---|---|---|
| | | LOSS | Ref | Zlib | Nei | Mink | Mink++ | **Our SMIA** |
| Pythia-12B | Wikipedia | 55.33 | 58.87 | 55.04 | 55.74 | 58.60 | 60.77 | **64.85** |
| | Github | 76.45 | 47.25 | 76.60 | 73.03 | 76.90 | 77.54 | **99.71** |
| | ArXiv | 48.66 | **57.63** | 47.14 | 51.83 | 49.91 | 53.12 | 54.45 |
| | PubMed | 53.20 | 56.73 | 51.86 | 57.77 | 53.28 | 55.66 | **68.39** |
| Pythia-6.9B | Wikipedia | 54.20 | 57.15 | 54.14 | 54.39 | 57.89 | 57.36 | **62.86** |
| | Github | 75.72 | 47.52 | 75.87 | 73.08 | 76.17 | 77.31 | **99.64** |
| | ArXiv | 48.28 | **55.96** | 46.79 | 51.79 | 48.87 | 51.34 | 54.01 |
| | PubMed | 52.18 | 52.02 | 51.05 | 56.71 | 52.10 | 53.82 | **61.90** |

members may have been seen during training, complicating the distinction between members and non-members (Duan et al., 2024).

**Larger models memorize more:** Another observation from Table 1, Table 2, Table 4, and Table 5 is that larger models exhibit greater memorization, consistent with findings from previous studies (Duan et al., 2024a; Carlini et al., 2022a; Nasr et al., 2023). For instance, for the $WT$ ($WC$) test datasets, SMIA achieves AUC-ROC scores of 67.39% (93.35%), 64.63% (92.11%), and 60.65% (89.97%) for Pythia 12B, 6.9B, and 2.7B, respectively. Similarly, SMIA achieves 59.71% (89.59%) and 58.92% (87.43%) on GPT-Neo 2.7B and 1.3B, respectively, for the $WT$ ($WC$) test datasets.

## 5 CONCLUSION

In this paper, we introduced the Semantic Membership Inference Attack (SMIA), which leverages the semantics of input texts and their perturbations to train a neural network for distinguishing members from non-members. We evaluated SMIA in two primary settings: (1) where the test member dataset exists verbatim in the training dataset of the target model, and (2) where the test member dataset is slightly modified through the addition, duplication, or deletion of a single word.

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

## A  Evaluation in Modified Settings

Existing MIAs against LLMs typically assess the membership status of texts that exist verbatim in the training data. However, in practical scenarios, member data might undergo slight modifications. An important application of MIAs is identifying the use of copyrighted content in training datasets. For instance, in the legal case involving the New York Times and OpenAI, the outputs of ChatGPT were found to be very similar to NYTimes articles, with only minor changes such as the addition or deletion of a single word (Grynbaum & Mac, 2023). This section explores the capability of SMIA to detect memberships even after such slight modifications.

To evaluate SMIA and other MIAs under these conditions, we generated three new test member datasets from our existing Wikipedia test member dataset ($D_{\text{te-m}}$) as follows (Figure 5 provides examples for each modification): **Duplication:** A random word in each member data point is duplicated. **Deletion:** A random word in each member data point is deleted. **Addition:** A mask placement is randomly added in each member data point, and the T5 model is used to fill the mask position, with only the first word of the T5 replacement being used. We just consider one word modification beacuse more than one word modification reuslts in a drastic drop of performance for all attacks.

Table 3 presents the AUC-ROC performance results of different MIAs and our SMIA under these slightly modified test member datasets. The table includes the best AUC-ROC values for Min-K and Min-K++ across different values of $K$. The results indicate that for the $WT$ non-member dataset, when a word is duplicated or added from the T5 output, the Ref attack outperforms SMIA. For instance, with Pythia-12B, the Ref attack achieves AUC-ROC scores of 57.88% and 57.95% after word duplication and addition from the T5 output, respectively, whereas SMIA achieves scores of 55.13% and 54.19% for the same settings. It is important to note that the Reference model is Pythia-1.4B, which shares the same architecture and training dataset (Pile) but with fewer parameters, a scenario that is less feasible in real-world applications. However, when a word is deleted, SMIA retains much of its efficacy, achieving an AUC-ROC of 62.47% compared to 58.25% for the Ref attack on the $WT$ non-member dataset. This indicates that SMIA is more sensitive to additions than deletions.

In scenarios involving the $WC$ non-member dataset, where non-members exhibit lower n-gram overlap with members, SMIA consistently outperforms other MIAs. For example, SMIA achieves AUC-ROC scores of 89.36% and 92.67% for word addition and deletion, respectively, while the Ref attack scores 66.50% and 66.84% for these modified member datasets.

Another key observation is that Min-K++ exhibits a greater decline in AUC-ROC than Min-K following modifications. For instance, on Pythia-12B with the $WC$ non-member dataset, Min-K++ AUC-ROC drops from 76.05% (no modification) to 69.07% (duplication), 70.81% (addition), and 69.87% (deletion). Conversely, Min-K AUC-ROC decreases from 76.05% (no modification) to 69.46% (duplication), 71.10% (addition), and 70.48% (deletion). This increased sensitivity of Min-K++ to modifications is due to its reliance on the average and variance of all vocabulary probabilities to normalize its scores, making it more susceptible to changes in these probabilities, thereby degrading performance.

## B  Existing MIAs against LLMs

**MIAs use cases:**  MIAs provide essential assessments in various domains. They are cornerstone for privacy auditing (Mireshghallah et al., 2022; Mattern et al., 2023), where they test whether LLMs leak sensitive information, thereby ensuring models do not memorize data beyond their learning scope. In the realm of machine unlearning Eldan & Russinovich (2023), MIAs are instrumental in verifying the efficacy of algorithms to comply with the right to be forgotten, as provided by privacy laws like the General Data Protection Regulation (GDPR) (Voigt & Von dem Bussche, 2017) and the California Consumer Privacy Act (CCPA) (Pardau, 2018). These attacks are also pivotal in copyright detection, pinpointing the unauthorized inclusion of copyrighted material in training

Table 3: Performance of various MIAs including our SMIA under different modification scenarios of the test member dataset ($D_{\text{te}}^m$). The table compares AUC-ROC scores when test members undergo word duplication, deletion, or addition using the T5 model. The results highlight the robustness of SMIA, especially against deletions and when non-members have lower n-gram overlap with members.

| Method | Modification | Pythia-12B | | Pythia-6.9B | |
|---|---|---|---|---|---|
| | | WT | WC | WT | WC |
| Duplication | Loss | 52.07 | 64.60 | 51.41 | 63.04 |
| | Ref | **57.88** | 66.48 | **55.94** | 62.72 |
| | Zlib | 51.87 | 63.99 | 51.23 | 62.44 |
| | Nei | 51.71 | 68.13 | 51.09 | 66.87 |
| | Mink | 51.93 | 69.46 | 51.10 | 67.45 |
| | Mink++ | 46.37 | 69.07 | 44.94 | 66.46 |
| | **Our SMIA** | 55.13 | **90.53** | 52.68 | **88.80** |
| Addition | Loss | 52.36 | 64.90 | 51.70 | 63.33 |
| | Ref | **57.95** | 66.55 | **56.05** | 62.84 |
| | Zlib | 52.31 | 64.47 | 51.65 | 62.93 |
| | Nei | 51.55 | 67.80 | 50.94 | 66.61 |
| | Mink | 52.60 | 71.10 | 51.75 | 69.02 |
| | Mink++ | 48.23 | 70.81 | 46.60 | 68.15 |
| | **Our SMIA** | 54.19 | **89.36** | 51.97 | **87.69** |
| Deletion | Loss | 51.83 | 64.28 | 51.19 | 62.74 |
| | Ref | 58.25 | 66.84 | 56.61 | 63.40 |
| | Zlib | 50.58 | 62.44 | 49.90 | 60.89 |
| | Nei | 54.55 | 70.65 | 53.99 | 69.50 |
| | Mink | 52.07 | 70.48 | 51.24 | 68.36 |
| | Mink++ | 47.46 | 69.87 | 46.04 | 67.30 |
| | **Our SMIA** | **62.47** | **92.67** | **60.39** | **91.37** |

datasets(Shi et al., 2023; Grynbaum & Mac, 2023). Furthermore, they aid in detecting data contamination – where specific task data might leak into a model's general training dataset (Wei et al., 2021; Chowdhery et al., 2023). Lastly, in the tuning the hyperparameters of differential privacy, MIAs provide insights for setting the $\epsilon$ parameter (i.e., the privacy budget), which dictates the trade-off between a model's performance and user privacy (Lowy et al., 2024; Bernau et al., 2019; Mireshghallah et al., 2022).

MIAs assign a membership score $A(x, T)$ to a given text input $x$ and a trained model $T(.)$. This score represents the likelihood that the text was part of the dataset on which $T(.)$ was trained. A threshold $\epsilon$ is then applied to this score to classify the text as a member if it is higher than $\epsilon$, and a non-member if it is lower. In this section, we provide the description of existing MIAs against LLMS.

**LOSS (Yeom et al., 2018):** The LOSS method utilizes the loss value of model $T(.)$ for the given text $x$ as the membership score; a lower loss suggests that the text was seen during training, so $A(x, T) = \ell(T, x)$.

**Ref (Carlini et al., 2021):** Calculating membership scores based solely on loss values often results in high false negative rates. To improve this, a difficulty calibration method can be employed to account for the intrinsic complexity of $x$. For example, repetitive or common phrases typically yield low loss values. One method of calibrating this input complexity is by using another LLM, $Ref(.)$, assumed to be trained on a similar data distribution. The membership score is then defined as the difference in loss values between the target and reference models, $A(x, T) = \ell(x, T) - \ell(x, Ref)$. Follwoing recent works (Shi et al., 2023; Zhang et al., 2024), we use smaller reference models, Pythia 1.4B and Pythia 70M, which are trained on the same dataset (Pile) and share a similar architecture with the Pythia target models.

**Zlib (Carlini et al., 2021):** Another method to calibrate the difficulty of a sample is by using its zlib compression size, where more complex sentences have higher compression sizes. The membership score is then calculated by normalizing the loss value by the zlib compression size, $A(x, T) = \frac{\ell(x,T)}{zlib(x)}$.

**Nei (Mattern et al., 2023; Mitchell et al., 2023):** This method perturbs the given text to calibrate its difficulty without the need for a reference model. Neighbors are created by masking random words and replacing them using a masking model like BERT (Devlin et al., 2018) or T5 (Raffel et al., 2020). If a model has seen a text during training, its loss value will generally be lower than the average of its neighbors. The membership score is the difference between the loss value of the original text and the average loss of its neighbors, $A(x, T) = \ell(x, T) - \frac{1}{n} \sum_{i \in [n]} \ell(\hat{x}_i, T)$, where in our experiments for each sample $n = 25$ neighbors are generated using a T5 model with 3B parameters.

**Min-K (Shi et al., 2023):** This attack hypothesizes that non-member samples often have more tokens assigned lower likelihoods. It first calculates the likelihood of each token as Min-K%$_{\text{token}}(x_t) = \log p(x_t|x_{<t})$, for each token $x_t$ given the prefix $x_{<t}$. The membership score is then calculated by averaging over the lowest $K\%$ of tokens with lower likelihood, $A(x, T) = \frac{1}{|\text{min-k\%}|} \sum_{x_i \in min-k\%} \text{Min-K\%}_{\text{token}}(x_t)$.

**Min-K++ (Zhang et al., 2024):** This method improves on Min-K by utilizing the insight that maximum likelihood training optimizes the Hessian trace of likelihood over the training data. It calculates a normalized score for each token $x_t$ given the prefix $x_{<t}$ as Min-K%++$_{\text{token}}(x_t) = \frac{\log p(x_t|x_{<t}) - \mu_{x_{<t}}}{\sigma_{x_{<t}}}$, where $\mu_{x_{<t}}$ is the mean log probability of the next token across the vocabulary, and $\sigma_{x_{<t}}$ is the standard deviation. The membership score is then aggregated by averaging the scores of the lowest $K\%$ tokens, $A(x, T) = \frac{1}{|\text{min-k\%++}|} \sum_{x_i \in min-k\%} \text{Min-K\%++}_{\text{token}}(x_t)$.

## C SMIA COST ESTIMATION

The cost estimation for deploying the SMIA involves several computational and resource considerations. Primarily, the cost is associated with generating neighbours, calculating embeddings, and evaluating loss values for the target model $T(.)$.

For each of the datasets, $D_{\text{tr-m}}$ (members) and $D_{\text{tr-n}}$ (non-members), consisting of $\beta$ data samples each, we generate $n$ neighbours per data item. Consequently, this results in a total of $2 \times n \times \beta$ neighbour generations. Assuming each operation has a fixed cost, with $c_N$ for generating a neighbour, $c_T$ for computing a loss value, and $c_E$ for calculating an embedding, the total cost for the feature collection phase can be approximated as: $2 \times (n \times \beta + 1) \times (c_N + c_E + c_T)$. In this estimation, the training of the neural network model $A(.)$ is considered negligible due to its relatively small size (few million parameters) and its architecture, which primarily consists of fully connected layers. Additionally, the costs associated with $c_T$ and $c_N$ are not significant in this context as they are incurred only during the inference phase. Thus, the predominant cost factor is $c_E$, the cost of embedding calculations.

In practical terms, for our experimental setup (Section 3.2) using the Wikipedia dataset as an example, we prepared a training set comprising 6,000 members and 6,000 non-members. With each data item generating $n = 25$ neighbours, the total number of data items requiring embedding calculations becomes: $6,000 + 6,000 + 150,000 + 150,000 = 312,000$ Each of these data items, on average, consists of 1052 characters (variable due to replacements made by the neighbour generation model), leading to a total of $312,000 \times 1052 = 328,224,000$ characters processed. These transactions are sent to the Cohere Embedding V3 model (Cohere, 2024) for embedding generation. The cost of processing these embeddings is measured in thousands of units. Hence, the total estimated cost for embedding processing is approximately: $32,822 \times \$0.001 = \$32.82$.

## D MISSING RESULTS

### D.1 TPR FOR LOW FPR

Table 4 and Table 5 show the TPR at 2%, 5% and 10% FPR for different baselines and our proposed SMIA by targeting different models using Wikipedia and MIMIR datasets.

Table 4: True Positive Rate (TPR) at 2%, 5% and 10% False Positive Rate (FPR) for various MIAs, including our SMIA, across different trained models (Pythia and GPT-Neo) using the Wikipedia dataset.

| Method | FPR | Pythia-12B | | Pythia-6.9B | | Pythia-2.7B | | GPT-Neo2.7B | | GPT-Neo1.3B | |
|---|---|---|---|---|---|---|---|---|---|---|---|
| | | WT | WC | WT | WC | WT | WC | WT | WC | WT | WC |
| LOSS | 2% | 2.1 | 12.2 | 2.6 | 11.9 | **3.1** | 9.4 | **2.8** | 9.2 | 2.2 | 8.7 |
| | 5% | 5.8 | 22.8 | 5.5 | 20.3 | 5.6 | 19.9 | 5.6 | 19.8 | 5.6 | 19.2 |
| | 10% | 11.1 | 32.1 | 10.9 | 29.8 | 10.2 | 28.6 | 10.0 | 27.6 | 9.8 | 25.6 |
| Ref (Pythia 70M) | 2% | 1.1 | 5.3 | 1.7 | 4.7 | 2.1 | 4.3 | 1.7 | 4.4 | 1.6 | 3.0 |
| | 5% | 6.7 | 8.7 | 6.2 | 9.1 | 6.4 | 8.4 | 5.8 | 9.0 | 5.6 | 7.8 |
| | 10% | 11.7 | 18.3 | 12.0 | 16.1 | 11.1 | 14.8 | 11.9 | 13.9 | 12.3 | 13.3 |
| Ref (Pythia 1.4B) | 2% | 2.0 | 5.7 | 2.1 | 6.3 | 2.8 | 4.1 | 2.3 | 2.7 | 1.7 | 0.7 |
| | 5% | 8.2 | 13.1 | 7.3 | 10.2 | 5.9 | 7.2 | 5.3 | 5.3 | 4.2 | 2.7 |
| | 10% | 16.5 | 21.3 | 14.7 | 17.7 | 10.7 | 13.7 | 11.4 | 10.8 | 10.0 | 8.2 |
| Zlib | 2% | 2.2 | 12.0 | 2.1 | 10.5 | 1.9 | 9.2 | 2.1 | 10.0 | 2.0 | 9.9 |
| | 5% | 5.5 | 22.7 | 5.9 | 22.4 | 5.2 | 19.8 | 6.0 | 18.7 | 5.9 | 16.1 |
| | 10% | 10.4 | 33.5 | 10.2 | 30.5 | 9.1 | 29.2 | 10.0 | 28.1 | 9.9 | 28.3 |
| Nei | 2% | 1.3 | 11.2 | 1.4 | 9.3 | 1.7 | 9.0 | 1.5 | 10.1 | 2.1 | 8.9 |
| | 5% | 4.3 | 19.2 | 4.5 | 18.9 | 5.2 | 18.8 | 5.3 | 18.2 | 5.5 | 16.1 |
| | 10% | 10.4 | 32.0 | 10.5 | 29.9 | 10.0 | 27.6 | 10.4 | 27.4 | 11.0 | 28.3 |
| Min-K ($K = 10\%$) | 2% | 1.8 | 17.9 | 1.9 | 18.3 | 1.9 | 14.1 | 1.6 | 15.9 | 1.4 | 14.2 |
| | 5% | 5.6 | 28.9 | 6.0 | 26.0 | 6.7 | 23.9 | 5.7 | 22.4 | 6.5 | 20.0 |
| | 10% | 13.3 | 41.7 | 13.7 | 36.7 | 12.3 | 33.8 | 13.2 | 31.7 | 11.6 | 28.9 |
| Min-K ($K = 20\%$) | 2% | 1.8 | 14.7 | 2.1 | 16.7 | 2.7 | 14.1 | 2.4 | 14.3 | 2.0 | 13.9 |
| | 5% | 5.6 | 27.0 | 5.4 | 25.9 | 5.8 | 23.6 | 5.7 | 23.3 | 5.9 | 21.9 |
| | 10% | 12.7 | 38.3 | 12.6 | 36.5 | 12.0 | 31.9 | 12.4 | 32.2 | 11.4 | 28.4 |
| Min-K ($K = 30\%$) | 2% | 2.1 | 14.2 | 2.5 | 14.6 | 2.8 | 12.1 | 2.5 | 12.1 | 2.1 | 11.0 |
| | 5% | 5.8 | 28.4 | 5.5 | 25.3 | 5.5 | 22.2 | 5.5 | 22.3 | 5.4 | 19.7 |
| | 10% | 12.6 | 37.7 | 12.5 | 33.2 | 12.4 | 32.9 | 12.1 | 30.6 | 11.3 | 28.5 |
| Min-K++ ($K = 10\%$) | 2% | 3.0 | 19.4 | 2.2 | 13.6 | 2.5 | 12.4 | 2.8 | 12.3 | 2.2 | 10.0 |
| | 5% | 6.1 | 29.6 | 6.8 | 26.0 | 6.6 | 23.9 | 5.3 | 22.0 | 5.2 | 19.1 |
| | 10% | 12.6 | 40.6 | 12.6 | 40.4 | 11.9 | 32.2 | 10.2 | 33.2 | 11.4 | 28.3 |
| Min-K++ ($K = 20\%$) | 2% | 2.8 | 21.2 | 2.0 | 17.4 | 2.3 | 16.7 | 2.7 | 12.9 | 2.0 | 11.7 |
| | 5% | 5.5 | 30.5 | 6.0 | 27.7 | 5.6 | 23.7 | **6.5** | 24.1 | 5.3 | 20.1 |
| | 10% | 12.2 | 43.7 | 12.0 | 38.7 | 12.2 | 34.8 | 10.2 | 34.0 | 10.9 | 29.6 |
| Min-K++ ($K = 30\%$) | 2% | 2.7 | 20.9 | 2.0 | 17.7 | 2.2 | 16.9 | 2.7 | 12.8 | 2.1 | 11.3 |
| | 5% | 5.4 | 31.4 | 5.8 | 27.5 | 5.7 | 24.8 | 6.4 | 24.6 | 4.7 | 20.2 |
| | 10% | 12.2 | 43.9 | 12.5 | 38.3 | 11.5 | 35.4 | 10.5 | 34.4 | 11.3 | 30.5 |
| Our SMIA | 2% | **3.8** | **46.2** | **3.1** | **41.6** | 2.4 | **35.1** | 1.8 | **32.9** | **2.8** | **25.2** |
| | 5% | **10.4** | **66.0** | **8.3** | **60.2** | **6.8** | **52.5** | 6.3 | **49.8** | **7.2** | **45.4** |
| | 10% | **20.6** | **79.3** | **18.1** | **75.4** | **15.0** | **67.6** | **14.4** | **67.9** | **14.9** | **60.5** |

## D.2 EFFECT OF DEDUPLICATION

Table 6 shows the AUC-ROC metric comparing different MIAs and our SMIA for deduped pythia models using Wikipedia dataset.

## D.3 EFFECFT OF NUMBER OF NEIGHBOURS

Table 7 presents the performance of our SMIA when varying the number of neighbors used during inference. The results indicate that a larger number of neighbors generally improves SMIA's performance. However, we have chosen to use 25 neighbors in our experiments, as increasing this

Table 5: True Positive Rate (TPR) at 2%, 5% and 10% False Positive Rate (FPR) for various MIAs, on datasets in MIMIR dataset (Duan et al., 2024) where members and non-members share less than 80% overlap in 13-gram.

| Target Model | Dataset | FPR | Method | | | | | | |
|---|---|---|---|---|---|---|---|---|---|
| | | | LOSS | Ref | Zlib | Nei | Mink | Mink++ | **Our SMIA** |
| Pythia-12B | Wikipedia | 2% | 4.04 | 1.01 | 5.78 | 2.89 | 4.62 | 2.89 | **10.40** |
| | | 5% | 10.98 | 5.78 | 9.24 | 5.78 | 10.40 | 7.51 | **16.18** |
| | | 10% | 16.18 | 10.40 | 13.29 | 12.13 | 19.07 | 17.34 | **23.69** |
| | Github | 2% | 40.46 | 6.35 | 39.30 | 15.60 | 40.46 | 38.72 | **95.95** |
| | | 5% | 46.24 | 13.87 | 46.24 | 27.74 | 45.66 | 43.93 | **98.84** |
| | | 10% | 50.28 | 19.65 | 54.91 | 39.88 | 50.28 | 51.44 | **100** |
| | Arxiv | 2% | 1.50 | 1.00 | 1.00 | 0.00 | 1.00 | **2.51** | 1.00 |
| | | 5% | 4.52 | 6.53 | 5.02 | 2.51 | 6.53 | **7.53** | 3.51 |
| | | 10% | 8.54 | **16.58** | 6.53 | 4.52 | 8.54 | 11.05 | 12.60 |
| | PubMed | 2% | 0.00 | 7.00 | 0.00 | 1.00 | 0.00 | 1.00 | **8.50** |
| | | 5% | 9.50 | 9.50 | 6.00 | 6.00 | 8.00 | 6.00 | **11.50** |
| | | 10% | 13.50 | 18.50 | 15.00 | 14.50 | 14.5 | 12.50 | **30.50** |
| Pythia-6.9B | Wikipedia | 2% | 5.20 | 0.00 | 7.51 | 3.46 | 6.35 | 5.78 | **8.67** |
| | | 5% | 12.13 | 3.46 | 9.82 | 5.20 | 10.98 | 10.40 | **14.45** |
| | | 10% | 15.60 | 6.35 | 14.45 | 12.13 | 19.65 | 17.91 | **19.65** |
| | Github | 2% | 34.68 | 7.51 | 37.52 | 19.07 | 36.41 | 32.36 | **97.10** |
| | | 5% | 41.61 | 9.24 | 42.77 | 28.32 | 41.04 | 43.35 | **98.84** |
| | | 10% | 47.97 | 19.65 | 51.44 | 44.50 | 45.08 | 49.71 | **100** |
| | Arxiv | 2% | 1.50 | **3.01** | 1.00 | 0.00 | 1.50 | 2.51 | 1.00 |
| | | 5% | 5.52 | **6.53** | 4.52 | 3.01 | 6.03 | 4.52 | 3.01 |
| | | 10% | 9.04 | 10.55 | 7.53 | 8.54 | 9.54 | **12.06** | 8.54 |
| | PubMed | 2% | 0.00 | **6.00** | 0.00 | 1.00 | 0.00 | 0.00 | 5.50 |
| | | 5% | 8.00 | 10.50 | 4.50 | 4.50 | 7.00 | 6.50 | **14.00** |
| | | 10% | 13.50 | 16.50 | 15.00 | 13.00 | 16.00 | 12.50 | **23.50** |

number further leads to additional computational demands without a corresponding improvement in performance.

### D.4 EFFECT OF SIZE OF TRAINING DATASET

Figure 3 illustrates the effect of using larger training datasets on the validation loss of the SMIA over 20 epochs. This figure displays the performance of the SMIA model on a validation dataset consisting of 1,000 members and 1,000 non-members, which are existing in the original Wikipedia portion of the Pile dataset (train and validation splits). In our experiments, we tested four different training sizes: 1,000 members + 1,000 non-members, 2,000 members + 2,000 non-members, 4,000 members + 4,000 non-members, and 6,000 members + 6,000 non-members. The results indicate that larger training datasets generally yield lower validation losses for the SMIA model. However, larger datasets require more computational effort as each member and non-member sample needs $n$ neighbors generated, followed by the calculation of embedding vectors and loss values for each neighbor. Due to computational resource limitations, we use a training size of 6,000 members + 6,000 non-members for all our experiments.

### D.5 SIMILARITY SCORES OF NEIGHBOURS

Figure 4 shows histogram of the similarity scores between members, non-members, and their 25 generated neighbors. These similarity scores are calculated using cosine similarity between the embedding vector of the original text and the embedding vectors of the neighbors. The dataset comprises 6,000 members and 6,000 non-members, resulting in 150,000 neighbors for each group. The histogram reveals that while most neighbors exhibit high similarity, there is a range of variability. Notably, even neighbors with lower similarity scores, such as around 70%, provide valuable data for training our SMIA. This diversity enables SMIA to more effectively distinguish membership under varying degrees of textual context changes.

Table 6: AUC-ROC performance metrics for various MIAs, including our SMIA, across different trained deduped Pythia models using the Wikipedia dataset.

| Method | Pythia-12B-Deduped | | Pythia-6.9B-Deduped | | Pythia-2.7B-Deduped | |
|---|---|---|---|---|---|---|
| | WT | WC | WT | WC | WT | WC |
| LOSS | 53.39 | 65.19 | 53.08 | 61.58 | 52.78 | 62.93 |
| Ref (Pythia 70M) | 50.24 | 54.85 | 49.71 | 53.89 | 48.92 | 51.90 |
| Ref (Pythia 1.4B) | 51.62 | 56.99 | 50.32 | 54.70 | 47.40 | 47.09 |
| Zlib | 52.81 | 64.31 | 52.55 | 63.64 | 52.23 | 62.08 |
| Nei | 53.93 | 69.93 | 53.63 | 69.17 | 53.07 | 68.16 |
| Min-K ($K = 10\%$) | 54.40 | 72.91 | 53.71 | 71.40 | 53.62 | 69.39 |
| Min-K ($K = 20\%$) | 54.25 | 70.83 | 53.77 | 69.80 | 53.41 | 67.95 |
| Min-K ($K = 30\%$) | 54.0 | 69.30 | 53.59 | 68.24 | 53.17 | 66.50 |
| Min-K++ ($K = 10\%$) | 52.84 | 74.48 | 52.13 | 72.92 | 51.32 | 69.90 |
| Min-K++ ($K = 20\%$) | 53.66 | 75.19 | 53.01 | 73.28 | 51.95 | 70.19 |
| Min-K++ ($K = 30\%$) | 54.00 | 74.62 | 53.28 | 72.74 | 52.11 | 69.49 |
| **Our SMIA** | **61.15** | **90.72** | **60.01** | **88.43** | **58.49** | **84.39** |

Table 7: AUC-ROC performance metrics of SMIA when different number of neighbors used in inference.

| Method | $n_{inf}$ | Pythia-12B | | Pythia-6.9B | | GPT-Neo-2.7B | |
|---|---|---|---|---|---|---|
| | | WT | WC | WT | WC | WT | WC |
| SMIA | 1 | 55.26 | 61.01 | 53.84 | 60.23 | 51.92 | 58.58 |
| | 2 | 58.48 | 70.60 | 56.41 | 68.82 | 53.26 | 66.18 |
| | 5 | 61.27 | 78.06 | 59.08 | 76.48 | 57.17 | 74.63 |
| | 15 | 65.63 | 87.15 | 62.62 | 85.46 | 58.86 | 82.60 |
| | 25 | **67.39** | **93.35** | **63.64** | **92.11** | **59.71** | **89.59** |

# E    MORE DETAILS ABOUT EXPERIMENT SETUP

## E.1    SMIA MODEL ARCHITECTURE

Table 8 shows the SMIA architecture with its layer sizes that we used in our experiments.

## E.2    METRICS

In our experiments, we employ following privacy metrics to evaluate the performance of our attacks:

**(1) Attack ROC curves:** The Receiver Operating Characteristic (ROC) curve illustrates the trade-off between the True Positive Rate (TPR) and the False Positive Rate (FPR) for the attacks. The FPR measures the proportion of non-member samples that are incorrectly classified as members, while the TPR represents the proportion of member samples that are correctly identified as members. We report the Area Under the ROC Curve (AUC-ROC) as an aggregate metric to assess the overall success of the attacks. AU-ROC is a threshold-independent metric, and it shows the probability that a positive instance (member) has higher score than a negative instance (non-member).

**(2) Attack TPR at low FPR:** This metric is crucial for determining the effectiveness of an attack at confidently identifying members of the training dataset without falsely classifying non-members as members. We focus on low FPR thresholds, specifically 2%, 5%, and 10%. For instance, the TPR

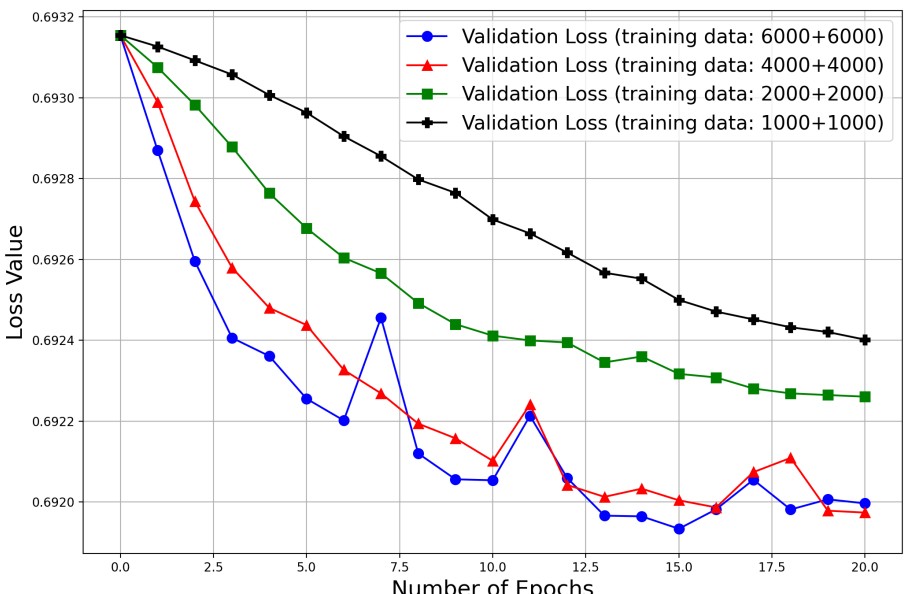

Figure 3: Effect of different training size on the validation loss of SMIA for 20 epochs.

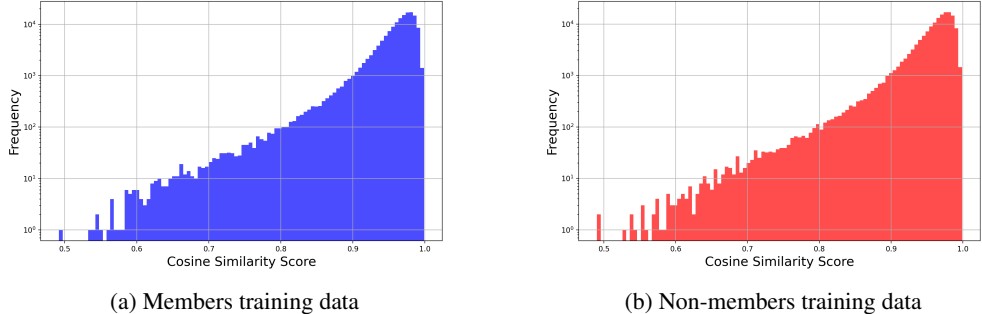

(a) Members training data                  (b) Non-members training data

Figure 4: Similarity scores of generated neighbors for our training datasets for member and non-member

at an FPR of 2% is calculated by setting the detection threshold so that only 2% of non-member samples are predicted as members.

### E.3 EXAMPLE OF MODIFIED TEXT

In Section 4, we introduce a modified evaluation setting where the member dataset undergoes various alterations. Figure 5 illustrates an example of a Wikipedia member sample undergoing different modifications: (a) shows the original sample, (b) shows a neighbor of the original created by replacing some words with outputs from a masking model, (c) shows modified sample by deleting a random word, (d) shows the modified sample by duplicating one word, and (e) shows the modified sample after adding one word using a T5 model.

Table 8: SMIA architecture with layer sizes

| Name | Layers | Details |
|---|---|---|
| Loss Component | 1 Fully Connected | FC(1,512)
Dropout (0.2)
ReLU activation |
| Embedding Component | 1 Fully Connected | FC(1024,512)
Dropout (0.2)
ReLU activation |
| Attack Encoding | 6 Fully Connected | FC(1024, 512), FC(512, 256),
FC(256, 128), FC(128, 64),
FC(64, 32) , FC(32, 1)
Dropout (0.2)
ReLU activation
Sigmoid |

Luis Manuel Blanco (born 13 December 1953) is an Argentine football coach, who currently manages Mons Calpe in the Gibraltar Premier Division. He was formerly the head coach of the Indonesia national team. His stay in Indonesia was brief, as he was replaced by Rahmad Darmawan after less than a month and no matches. ....

(a) Input sample (Original - with no modifiation)

Luis Manuel Blanco (born 13 December 1953) is an Argentine football manager , who currently manages Mons Calpe in the Gibraltar Premier League. Blanco was formerly the head coach of the Indonesia national team. Blanco's stay in Indonesia was brief, as he was dismissed at Indonesia by the Football Association of Indonesia after less than a month and a half, having failed to win any Indonesia national matches. ...

(b) Neighbor sample by substituting random words with a masking model output

Luis Manuel Blanco (born 13 December 1953) is an Argentine football coach, who currently manages Mons Calpe in the Gibraltar Premier Division. He was formerly the head coach of the Indonesia national team. His stay in Indonesia was brief, as he was replaced by Rahmad Darmawan after less than a month and no matches. ....

(c) Deletion modification by removing a random word

Luis Manuel Blanco (born 13 December 1953) is an Argentine football coach, who currently manages Mons Calpe in the Gibraltar Premier Premier Division. He was formerly the head coach of the Indonesia national team. His stay in Indonesia was brief, as he was replaced by Rahmad Darmawan after less than a month and no matches. ....

(d) Duplication modification by duplicating a random word

Luis Manuel Blanco (born 13 December 1953) is an Argentine football coach, who currently manages Mons Calpe in the Gibraltar Premier Division. He was formerly the head coach of the Indonesia national team. His stay in Indonesia where was brief, as he was replaced by Rahmad Darmawan after less than a month and no matches. ....

(e) Addition modification by adding the first word of a masking model for a random mask token

Figure 5: An example for input sample and different modifications.

## E.4 SMIA Hyperparameters

To construct our neighbor datasets, we generate $n = 25$ neighbors for each data point. Table 8 details the architecture of the SMIA model used across all experiments. We employ the Adam optimizer to train the network on our training data over 20 epochs. The batch size is set to 4, meaning each batch contains neighbors of 2 members and 2 non-members, totaling 50 neighbors for members and 50 neighbors for non-members, thus including 100 neighbors per batch. For regular experiments, we use a learning rate of $5 \times 10^{-6}$. However, for modified evaluations, which include duplication,

addition, and deletion scenarios, we adjust the learning rate to $1 \times 10^{-6}$. In all of our experiments, we report the AUC-ROC or TPR of the epoch that results in lowest loss on validation dataset.

### E.5 COMPUTE RESOURCES

For the majority of our experiments, we utilize a single H100 GPU with one core. It is important to note that we do not train or fine-tune any LLMs during our experiments; we operate in inference mode using pre-trained models such as the T5 masking model and various models from the Pythia family. Generating $n = 25$ neighbors for a dataset of 1,000 texts required approximately 16 hours of compute time. For the task of calculating embedding vectors, we employed the Cohere Embedding V3 model, which is provided as a cloud service. The computation of loss values for the target model was also minimal, taking only a few minutes for the a dataset of 1,000 examples. Finally, training the SMIA model was notably rapid, owing to its relatively small size of only a few million parameters. The entire training process, after having all the input features for training data, was completed in less than 10 minutes over 20 epochs.

