# OpenReview forum: "Semantic Membership Inference Attack against Large Language Models"
_NeurIPS.cc/2024/Workshop/SafeGenAi — SafeGenAi Poster_

### Official Review · Reviewer_8gnY · 2024-10-08
**Reliance on Grey-box Access**

**Rating:** 6
**Confidence:** 4

**Review:**

This paper introduces the Semantic Membership Inference Attack (SMIA), an advanced version of traditional Membership Inference Attacks (MIAs) designed to target large language models (LLMs). SMIA leverages the semantic content of inputs and their perturbations to enhance inference accuracy, distinguishing whether specific data points were included in the model’s training set. The paper presents a neural network-based method to assess a model’s behavior when presented with semantically perturbed inputs, improving upon existing MIA methods. Empirical evaluations, particularly on models like Pythia and GPT-Neo, demonstrate that SMIA outperforms existing approaches in detecting training data through semantic understanding, showcasing high AUC-ROC performance and TPR at low FPR settings.

Limitations:
1. SMIA assumes access to loss values or log probabilities of the target model, which may not always be feasible in real-world scenarios. This limitation is critical because, in practical deployments, adversaries often lack access to model internals, such as loss values or probabilities. Without such access, SMIA's real-world utility diminishes significantly, making this limitation highly consequential.
2. The performance of SMIA relies heavily on the generation and evaluation of a large number of perturbed neighbors, which may become computationally expensive for larger datasets or models. The need to generate numerous semantic neighbors and calculate embedding differences makes SMIA computationally demanding, especially when applied to large-scale models. This hinders its practical scalability in high-throughput environments.

---

### Official Review · Reviewer_PuTw · 2024-10-09

**Rating:** 7
**Confidence:** 4

**Review:**

**Summary**

The authors introduce Semantic Membership Attack (SMIA) that enhances MIA performance by leveraging semantic content of inputs and their perturbations. The paper provides an extensive suite of experiments to test performance of various other MIAs and also provides a good cost estimate.

Overall, good paper accept.

**Pros**

- The paper is well-structured and flows logically
- They also conduct extensive evaluation of several Membership Inference Attack (MIA) approaches applied to Large Language Models (LLMs).
- The SMIA method for detecting members versus non-members is relatively lightweight

**Cons**

- Generally, a frequent overlap is found between members and non-members datasets, although the authors have taken precautions using deduplication techniques and using the Wikipedia cutoff dataset the explanation is not precise enough. Since embeddings are already being computed for the members and non-members it would be good if Authors can compute score such as MAUVE mentioned in Questions.
- The evaluation is limited to 130-150 words limit, MIAs performance can vary based on the word length and testing attacks around different word limit short, medium and long is expected experiment setup in common literature which is missing here. The authors have mentioned about copyrights in modified settings and generally copyright infringements happen for larger content.

**Questions to Authors**

- Did the authors explore additional metrics beyond n-grams to assess the overlap between members and non-members, such as the MAUVE score? If so, which other metrics might be relevant for this analysis?
- I strongly suggest the authors to evaluate their work on the WikiMIA dataset which serves as a benchmark dataset for MIA attacks it is introduced in one of the reference papers : paper, dataset